# Multiway Information Interaction Measures Reveal Redundant Coding in Biological and Artificial Neural Networks

## Abstract

The investigation of information interaction in complex biological and deep neural networks is a significant research problem in neuroscience, deep learning, and information theory. However, there is limited research assessing multiway information interactions in the brain and deep neural networks, particularly regarding how information changes in each layer of neural networks. In this study, we explore redundancy representation in biological and standard deep convolutional neural networks. Firstly, we demonstrate that multiway information interaction, it can be quantified using total correlation, which indeed take precedence over pairwise metrics. Secondly, we propose that the data processing inequality holds in both biological vision and deep neural networks, and it confirms parallel information processing in the visual brain and deep neural networks. Furthermore, we observe that redundant information dominates in the early layers of the visual brain, while synergistic features gradually emerge in both the deeper layers of the visual cortex and deep neural networks, suggesting that general convolutional neural nets exhibit behavior similar to sensory information encoding in the primate visual brain.

## 1 Introduction

Understanding the black box of the human brain and deep learning is a key and pressing scientific question, and quantifying information representation is one way to explore it, alongside visualizing hidden feature representations in both the brain and deep neural networks (DNNs) Mahendran & Vedaldi (2014); Yosinski et al. (2015). Moreover, it also provides us with a way to align brain and deep neural network functions, helping us to better optimize deep neural networks, and at the same time, aiding in our understanding of brain functions Wang et al. (2023); Li et al. (2022a); Lindsay (2020); St-Yves et al. (2023).

Information theory offers a promising opportunity to achieve this goal, with a substantial body of related studies already investigating it Dimitrov et al. (2011); Cover & Thomas (2006). One classical example in deep neural networks is the autoencoder, which extensively explores concepts like information compression, the information bottleneck, and decompression. Tishby's work, for instance, explores changes in mutual information during autoencoder training, thus shedding light on the inner workings of DNNs from an information-theoretical perspective Tishby & Zaslavsky (2015); Yu & Principe (2019).

There indeed exists a substantial body of research on measuring information alterations in DNNs Painsky & Tishby (2018). However, most of these studies only utilize pairwise metrics such as mutual information for quantification. As we know, mutual information can only measure pairwise linear and nonlinear information, limiting its applicability because most signals are high-dimensional and their relationships inside extend beyond pairwise Ferenci & Kovacs (2014); Li et al. (2022b; 2023); Laparra et al. (2011). Therefore, in this study, we explored both pairwise and beyond-pairwise metrics using mutual information and total correlation to quantify information, as shown in Fig.1, and demonstrated that these measures are necessary for capturing high-order or multiway interactions in complex biological and deep neural networks.

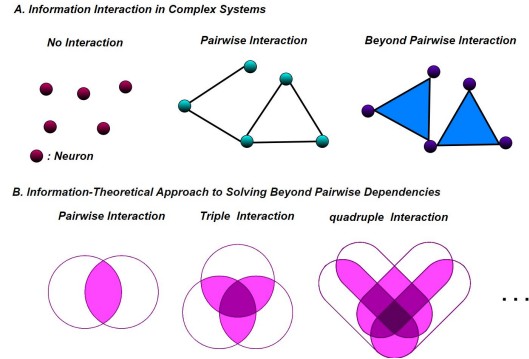

Furthermore, there has been limited exploration of redundancy reduction using total correlation in both the brain and DNNs. Therefore, our primary objectives include quantifying redundant information in both the brain and DNNs, and attempting to capture hidden high-order interactions using beyond pairwise metrics. Before quantifying independent redundancy in these systems, we first demonstrated that relying solely on pairwise metrics is insufficient to capture high-order interactions in complex systems. Subsequently, we applied this approach to real brain signals and DNNs, uncovering both functional similarities and differences between deep neural networks and the visual brain.

Figure 1: In complex systems, information interaction goes beyond pairwise connections. Some neurons act independently, while others engage in pairwise or higher-order interactions (**A**). Total correlation is one metric that captures these higher-order dependencies (**B**).

Our main contributions can be summarized as follows:

- Introduced and applied beyond-pairwise information-theoretic metrics (e.g., total correlation) to quantify high-order interactions in both biological and deep neural networks.
- Demonstrated the limitations of pairwise metrics by showing their insensitivity to capturing high-order dependencies.
- Revealed functional parallels and distinctions between deep neural networks and the visual brain through high-order redundancy analysis, offering new insights into their internal representations.

## 2 THEORETICAL ANALYSIS

### 2.1 TOTAL CORRELATION

#### 2.1.1 MULTI-WAY RELATIONSHIP MEASURES WITH TOTAL CORRELATION

The Total Correlation ($TC$) describe the dependence among $n$ variables $(X^1, \cdots, X^n)$ and can be considered as a non-negative generalization of the concept of mutual information (**Appendix A1**) from two parties to $n$ parties. Let the definition of $TC$, as introduced by Watanabe Watanabe (1960), be denoted by:

$$TC\left(X^1, \cdots, X^n\right) = \sum_{i=1}^{n} h\left(X^i\right) - h\left(X^1, \cdots, X^n\right) \tag{1}$$

where $h\left(X^i\right)$ is marginal entropy, and $h\left(X^1, \cdots, X^n\right)$ is joint entropy. The TC will be equal to mutual information if we only have two variables, and $TC$ will be zero if all variables are totally independent.

### 2.2 TOTAL CORRELATION ESTIMATE WITH ROTATION-BASED ITERATIVE GAUSSIANIZATION

To quantify interactions beyond pairwise relationships, we proposed using Rotation-Based Iterative Gaussianization (RBIG) as a method for estimating $TC$ to capture higher-order interaction information in both brain networks and deep neural networks. The RBIG was proposed as a method for probability density function (PDF) estimation Laparra et al. (2011) and the estimation of information theoretic measures Laparra et al. (2025).

The RBIG is a cascade of $L$ *nonlinear+linear* layers, and the $l$-th layer is made of marginal Gaussianizations, $\Psi^{(l)}(\boldsymbol{x}^{(l)})$, followed by a rotation, $R^{(l)}$. Each of such layers is applied on the output of the previous layer (as shown in Fig.2):

$$\boldsymbol{x}^{(l+1)} = R^{(l)} \cdot \Psi^{(l)}(\boldsymbol{x}^{(l)}) \tag{2}$$

For a big enough number of layers, this invertible architecture is able to transform any input PDF, $p(\boldsymbol{x}^{(0)})$, into a zero-mean unit-covariance multivariate Gaussian even if the chosen rotations are random Laparra et al. (2011). $TC$ describes the redundancy within a vector, i.e., the information shared by the univariate variables Watanabe (1960); Studený & Vejnarová (1998). Note that strong relations between variables indicates a rich structure in the data.

Density destruction together with differentiability is useful to estimate the $TC$ within a vector $TC(\boldsymbol{x}^{(0)})$. Imagine that the considered RBIG transforms the PDF of the input $\boldsymbol{x}^{(0)}$ into a Gaussian through the application of $L$ layers ($L$ individual transforms). As the redundancy of the Gaussianized signal, $g_{\boldsymbol{x}}(\boldsymbol{x}^{(0)}) = \boldsymbol{x}^{(L)}$, is zero, the redundancy of the original signal, $TC(\boldsymbol{x}^{(0)})$, should correspond to the cumulative sum of the individual variations, $\Delta TC^{(l)}$ with $l = 1, \ldots L$, that take place along the $L$ layers of RBIG, while converting the original variable $\boldsymbol{x}$, into the Gaussianized variable $g_{\boldsymbol{x}}(\boldsymbol{x})$. Interestingly, the individual variation in each RBIG layer only depends on (easy to compute) univariate negentropies, therefore, after the $L$ layers of RBIG, the $TC$ is Laparra et al. (2011):

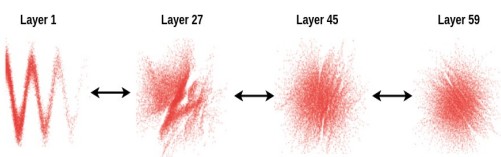

Figure 2: To transform non-Gaussian distributed data into a Gaussian distribution, we apply the RBIG method. The image on the left illustrates the data after the first layer, where marginal Gaussianization and rotation are applied. By the 59th layer, the data closely approximates a Gaussian distribution. Notably, this entire transformation process is reversible, meaning that data can be mapped back from the Gaussian distribution to its original non-Gaussian form.

$$TC(\boldsymbol{x}) = \sum_{l=1}^{L} \Delta T(\boldsymbol{x}^{(l-1)}, \boldsymbol{x}^{(l)}) = \sum_{l=1}^{L} J_m(\boldsymbol{x}^{(l)}) \tag{3}$$

where the marginal negentropy of a $d$-dimensional random vector is given by a set of $d$ univariate divergences $J_m(\boldsymbol{v}) = \sum_{i=1}^{d} D_{\mathrm{KL}}(p(v_i) | \mathcal{N}(0,1))$. Therefore, using RBIG, the challenging problem of estimating one $d$-dimensional joint PDF to compute $TC(\boldsymbol{x})$ reduces to solve $d \times L$ univariate problems. RBIG estimate of $TC$ has been shown to be better than previously reported estimates of $TC$ Laparra et al. (2025), and it has been successfully used in neuroscience to quantify the redundancy reduction over the visual pathway Gomez-Villa et al. (2020); Malo (2019); Li et al. (2023).

### 2.3 DATA PROCESSING INEQUALITY IN THE PHYSICAL COMPLEX SYSTEM

The Data Processing Inequality (DPI) is a fundamental principle in information theory, stating that information cannot be increased through operations conducted along the transmission path Cover & Thomas (2006). If we assume that $X^i - X^j - X^k$ is a Markov chain, then we must have:

$$I\left(X^i, X^j\right) \geq I\left(X^i, X^k\right) I\left(X^j, X^k\right)$$
$$\geq I\left(X^i, X^k\right) \tag{4}$$

Figure 3: Simulated functional connectivity between two nodes. Each panel shows how the signal in $x_b$ depends on $x_a$ under varying conditions: connectivity $c$ (left), non-shared signal weight $\gamma_a$ (center), and noise level $\sigma_{nb}$ (right). Rows correspond to different nonlinearities: linear (top), compressive (middle), and accelerating (bottom). Red dots represent samples from the baseline condition; other points show the effect of increasing the corresponding parameter.

Here we have that mutual information cannot increase along the chain, and it's the core principle of DPI.

## 3 EXPERIMENTS AND RESULTS

### 3.1 TOTAL CORRELATION IS MORE SENSITIVE THAN PAIRWISE METRICS

Following the synthetic experiments in Cole et al. (2015), we consider that the signal in two nodes $x_a$ and $x_b$ could come from a shared common $s_{ab}$, two independent sources, $s_a$ and $s_b$, and two noise sources, $n_a$ and $n_b$ (all of them Gaussian). In order to make a clearer difference with 2nd order correlation, lets expand the all-Gaussian scenario in Cole et al. (2015) by applying an arbitrary nonlinearity, $f(.)$, to one of the signals (e.g. the one in region $x_b$). In that case, the signals in $x_a$ and $x_b$ are:

$$x_a = c\,s_{ab} + \gamma_a\,s_a + n_a \tag{5}$$
$$x_b = f(\,c\,s_{ab} + \gamma_b\,s_b + n_b\,)$$

Where $f(.)$ could be a saturating or accelerating exponential for the sake of illustration, here we used divisive normalization Lyu & Simoncelli (2008); Burg et al. (2021) to control the saturating or accelerating exponential for generating multidimensional signals, and it can be formulated as follows:

$$f(\mathbf{e}) = \text{sign}(\mathbf{e}) \cdot \kappa \cdot \frac{|\mathbf{e}|^\gamma}{b + uH \cdot |\mathbf{e}|^\gamma} \tag{6}$$

Here $\mathbf{e} = c\,s_{ab} + \gamma_b\,s_b + n_b$ , and the matrix $H$ in the denominator represents the interaction between the neurons of the previous cortical layer $\mathbf{e}$. Here, $\gamma$, $u$, $b$, and $\kappa$ are hyperparameters, and we use the same parameters as in the previous study Li et al. (2023).

The weight of the common signal is $c$ and the weights of the independent signals are $\gamma_a$ and $\gamma_b$, and all the sources are Gaussian: $s_{ab} \sim \mathcal{N}(0, \sigma_{ab})$, $s_a \sim \mathcal{N}(0, \sigma_a)$, $s_b \sim \mathcal{N}(0, \sigma_b)$, $n_a \sim \mathcal{N}(0, \sigma_{na})$, and $n_b \sim \mathcal{N}(0, \sigma_{nb})$. In this setting, *connectivity* refers to the weight of the shared signal, denoted as the factor $c$. Here, we generate a multidimensional signal with $d = 9$, and a sample size of $5 \times 10^3$, as depicted in Fig.3.

In an all-Gaussian scenario, the theoretical $TC$ (information shared by the elements of $\boldsymbol{x} = [x_a\ x_b]^\top$) is easy to obtain Laparra et al. (2025). Nevertheless, according to Cardoso (2003), when dealing with Gaussian PDFs the $TC$ is fully determined by the 2nd order correlation because in general:

$$TC = C + J - J_m$$

Where: 1) $C$ is a descriptor of the 2nd order correlation defined from the covariance matrix $\Sigma$. Specifically, it is defined as $C = \frac{1}{2}\,log(\frac{\prod_i \Sigma_{ii}}{|\Sigma|})$. Which, in the two-dimensional case, reduces to the Pearson correlation: $C = -\frac{1}{2}log(1 - \rho^2)$.

2) $J$ is the so-called global *non-Gaussianity*, and it is given by the Kullback Leibler divergence of the data with regard to the best-fitting Gaussian.

3) $J_m$ is the marginal *non-Gaussianity*, given by the sum of univariate non-Gaussianities.

And therefore, in a particular all-Gaussian scenario $J = J_m = 0$, and hence, $TC = C$.

In our expanded setting $TC \neq C$. However, $TC$ is still easy to compute since $TC$ is invariant under dimension-wise transforms such as $f(\cdot)$ Kraskov et al. (2004). In that case, we still can apply the Gaussian expression for $T$ Laparra et al. (2025) despite we applied $f(\cdot)$, and hence we have:

$$TC(\boldsymbol{x}) = \sum_{i=1}^{2} log(\Sigma_{ii}) - \frac{1}{2}log|\Sigma| \tag{7}$$

where $\Sigma$ is the covariance matrix of $\boldsymbol{x}$:

$$\Sigma = \begin{pmatrix} c^2\,\sigma_{ab}^2 + \gamma_a^2\sigma_a^2 + \sigma_{na}^2 & c^2\,\sigma_{ab}^2 \\ c^2\,\sigma_{ab}^2 & c^2\,\sigma_{ab}^2 + \gamma_b^2\sigma_b^2 + \sigma_{nb}^2 \end{pmatrix}$$

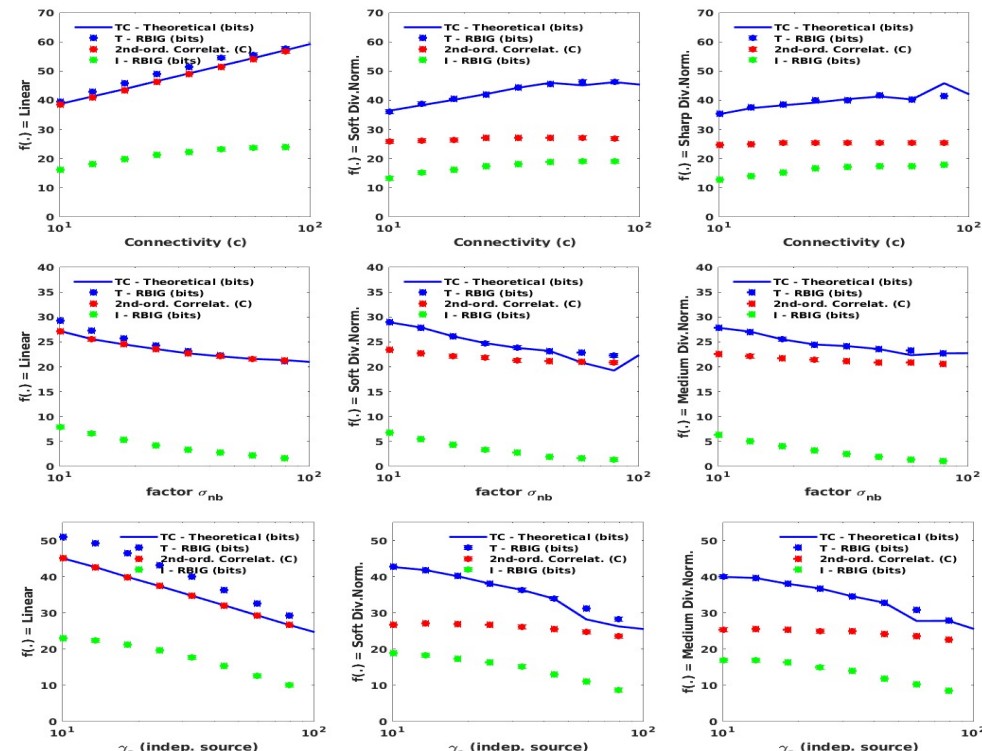

Figure 4: Sensitivity of correlation, $MI$, and $TC$ to connectivity. Here shows the dependence of $TC$ (theoretical and empirical) measured in *bits*, $MI$ measured in *bits*, and 2nd-order correlation obtained from samples with different connectivities, $c$ (first row), different amounts of noise, $\sigma_{nb}$ (second row), different weights of the non-shared signal, and $\gamma_a$ (third row). In each row, display the measures for different nonlinearities: linear case, soft nonlinearity, and accelerating nonlinearity.

and $\Sigma_i$ refer to the standard deviation in each dimension (i.e. the elements of the diagonal of $\Sigma$). So in summary, $TC$ can be analytically related to the connectivity $c$ which is in the elements of the covariance. Note that variations in the connectivity (shared source), noise and the weights of the non-shared source induce variations in $TC$.

In this controlled setting we may generate samples for a range of the connectivity parameters. Then, we may compare the quality of the descriptors of the connectivity by plotting the relation between the actual parameter and the value of the descriptor obtained from the samples. Fig.4 show the sensitivity of correlation, mutual information $MI$, and $TC$ to the variation of the connectivity.

Saturation of the correlation for a wide range of connectivity in multidimensional signal (as opposed to the monotonic behavior of $TC$) implies that $TC$ is way more appropriate than conventional pairwise metrics. Finally, note that in the linear case $TC$ can be successfully derived from $\rho$, and hence, as discussed above following Cardoso (2003), $TC$ has no advantage over 2nd order correlation. However, as we extended the experiments in Cole et al. (2015) by considering nonlinearities, in these more challenging situations the estimations from $MI$ and correlation are incorrect and have less sensitivity to connectivity than $TC$ since they saturate.

### 3.2 QUANTIFYING INFORMATION USING TOTAL CORRELATION IN REAL BIOLOGICAL VISION AND DEEP NETS

#### 3.2.1 BIOLOGICAL VISION AND DEEP NEURAL NETWORKS

To compare the flow of information in artificial neural networks with that of the biological visual system, as shown in Fig.5. We used data from the Algonauts Project 2021 Challenge Cichy et al.

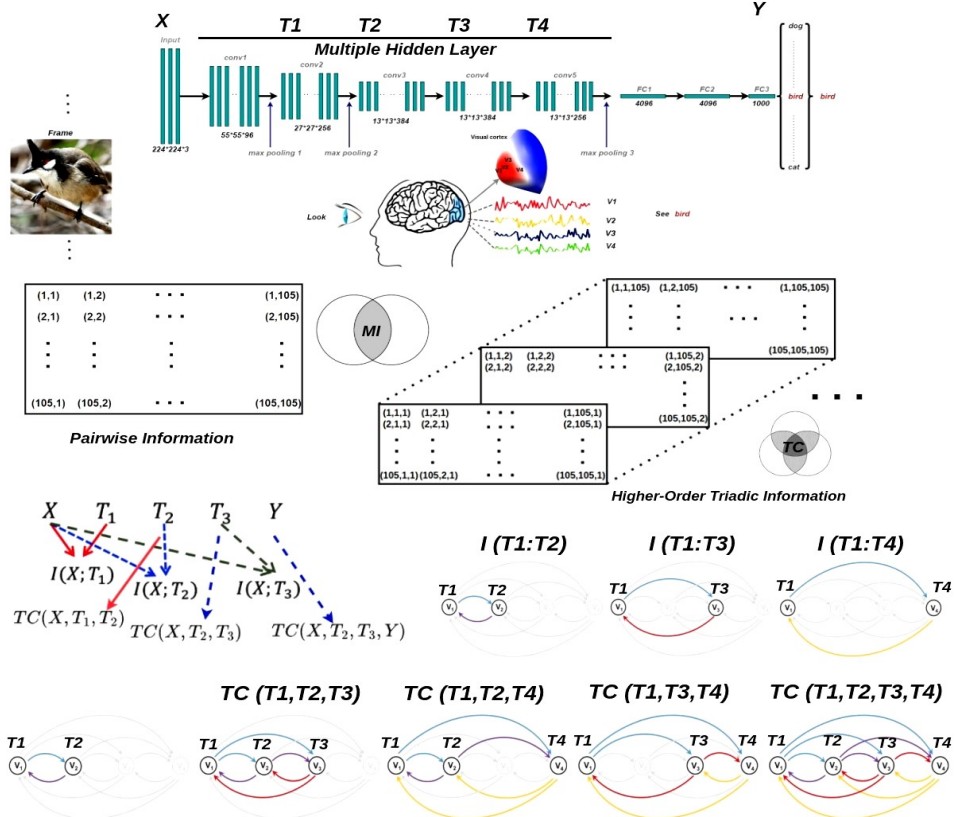

Figure 5: Information representation in biological visual systems and deep neural networks. The architecture of AlexNet and the visual brain both involve learning features at each layer of the neural network by inputting an video into the network. In AlexNet, the convolutional layers are labeled in each layer of the neural network, with max-pooling located at the bottom of the network. The final vector corresponds to the fully connected layer with 1000 classes, where the class with the highest probability is assigned as the correct prediction. In the visual brain, natural image processing occurs along the visual pathway, ultimately resulting in the perception of stimuli such as a bird. Then, pairwise interactions (using $MI$) and higher-order interactions (e.g., triple interactions using $TC$) will be quantified within both the visual brain and deep neural networks.

(2021) (**Appendix A2**). Specifically, we employed a pre-trained deep convolutional neural network (CNN), *AlexNet*, to extract hierarchical visual features from natural video stimuli (**Appendix A3**).

### 3.2.2 DATA PROCESSING INEQUALITY IN REAL BIOLOGICAL VISION AND DEEP NETS

We conducted separate examinations of information loss in biological and artificial neural networks, comparing the flow of information in pre-trained AlexNet with that in the human visual system (V1-V2-V3-V4) using the same stimuli. This comparison allows us to analyze information loss in deep neural networks and biological vision pathways from an information-theoretical perspective.

Considering that the internal representation may contain some amount of noise, we adopt the assumption, following Malo (2019), that this noise has an amplitude equal to 5% of the amplitude of the responses in each layer. Noise significantly impacts information loss because, in the transformation from $x$ to $y$, the information about the input $x$ conveyed by the noisy response $y$ is discussed in Malo (2019) as follows:

$$I(\boldsymbol{x}, \boldsymbol{y}) = \sum_i h(y_i) - T(\boldsymbol{y}) - h(\boldsymbol{n}) \tag{8}$$

where $h(\cdot)$ stands for the (univariate or joint) entropy of the corresponding variables, $T(\boldsymbol{y})$ refers to $TC$ of $\boldsymbol{y}$, and $h(\boldsymbol{n})$ indicates the joint entropy of the noise in the response. When constrained by an energy budget, the entropy of the response cannot increase without limit. Therefore, both redundancy $T$ and noise $\boldsymbol{n}$ play roles in the loss of information.

The information representation was estimated based on $MI$ for real biological visual and deep neural networks using Eq.8. First and most important, we see that DPI holds in biological vision (i.e., V1-V2-V3-V4: $2.3 \rightarrow 1.4 \rightarrow 1.0 \rightarrow 0.7$, V2-V3-V4: $2.0 \rightarrow 1.3 \rightarrow 0.7$, V3-V4: $1.7 \rightarrow 0.8$) (refer to Table.1) and also in deep neural networks (i.e., L1-L2-L3-L4: $2.3 \rightarrow 1.3 \rightarrow 0.8 \rightarrow 0.6$, L2-L3-L4: $1.9 \rightarrow 1.3 \rightarrow 0.6$, L3-L4: $2.0 \rightarrow 1.8$) (refer to Table.2).

| $\mathbf{I(V_i, V_j)}$ (in *bits/visual region*) | $V_1$ | $V_2$ | $V_3$ | $V_4$ |
|---|---|---|---|---|
| $V_1$ | $\mathbf{2.3} \pm 0.3$ | $\mathbf{1.4} \pm 0.4$ | $\mathbf{1.0} \pm 0.2$ | $\mathbf{0.7} \pm 0.2$ |
| $V_2$ | $1.4$ | $\mathbf{2.0} \pm 0.3$ | $\mathbf{1.3} \pm 0.2$ | $\mathbf{0.7} \pm 0.1$ |
| $V_3$ | $1.0$ | $1.3$ | $\mathbf{1.7} \pm 0.3$ | $\mathbf{0.8} \pm 0.2$ |
| $V_4$ | $0.7$ | $0.7$ | $0.8$ | $\mathbf{2.2} \pm 0.3$ |

| $\mathbf{I(L_i, L_j)}$ (in *bits/layer*) | $L_1$ | $L_2$ | $L_3$ | $L_4$ |
|---|---|---|---|---|
| $L_1$ | $\mathbf{2.3} \pm 0.3$ | $\mathbf{1.3} \pm 0.3$ | $\mathbf{0.8} \pm 0.1$ | $\mathbf{0.6} \pm 0.2$ |
| $L_2$ | $1.3$ | $\mathbf{1.9} \pm 0.1$ | $\mathbf{1.3} \pm 0.3$ | $\mathbf{0.6} \pm 0.1$ |
| $L_3$ | $0.8$ | $1.3$ | $\mathbf{2.0} \pm 0.2$ | $\mathbf{1.8} \pm 0.1$ |
| $L_4$ | $0.6$ | $0.6$ | $1.8$ | $\mathbf{2.0} \pm 0.2$ |

Table 1: The $MI$ in biological visual pathways. $\mathbf{MI(V_i, V_j)}$ between pairs of areas.

Table 2: The $MI$ in artificial neural networks (AlexNet). $\mathbf{MI(L_i, L_j)}$ between pairs of layers.

Furthermore, both biological and deep neural networks adhere to the DPI, and it supports that information is efficiently transmitted in parallel within both the visual brain and deep networks, as illustrated in Fig.6. This observation aligns with the *efficient coding hypothesis*, which posits that neurons encode information as compactly as possible to maximize resource efficiency, avoiding redundancy between neurons Barlow (1959; 2001).

### 3.2.3 REDUNDANCY AND SYNERGY IN REAL BIOLOGICAL VISION AND DEEP NETS

To further access redundancy and synergy distribution in biological and deep neural networks, we applied $TC$ to capture multi-neural information interaction in biological and deep neural networks. We observed some similarities and differences between biological vision and deep neural networks. Firstly, we noticed that redundant information decreases in the early visual system ($3.6 \rightarrow 3.2 \rightarrow 3.0$), but the other way around in deep neural networks ($3.6 \rightarrow 5.6 \rightarrow 6.0$) (refer to Table.3).

| $\mathbf{T(V_i)}$ (in *bits/visual region*) | $V_1$ | $V_2$ | $V_3$ | $V_4$ |
|---|---|---|---|---|
| | $3.5 \pm 0.3$ | $3.2 \pm 0.3$ | $3.0 \pm 0.3$ | $3.4 \pm 0.2$ |

| $\mathbf{T(L_i)}$ (in *bits/layer*) | $L_1$ | $L_2$ | $L_3$ | $L_4$ |
|---|---|---|---|---|
| | $3.6 \pm 0.1$ | $5.6 \pm 0.3$ | $6.0 \pm 0.3$ | $5.8 \pm 0.2$ |

Table 3: The $TC$ in biological visual pathways and deep neural networks. $\mathbf{TC(V_i)}$ in each visual area, and $\mathbf{TC(L_i)}$ in each layer.

| $\mathbf{\Delta T(v_i)}$ (in %) | $V_1$ | $V_2$ | $V_3$ | $V_4$ |
|---|---|---|---|---|
| | $0$ | $-8$ | $-15$ | $-1$ |

| $\mathbf{\Delta T(L_i)}$ (in %) | $L_1$ | $L_2$ | $L_3$ | $L_4$ |
|---|---|---|---|---|
| | $0$ | $56$ | $67$ | $61$ |

Table 4: Variations of $TC$ (in % with regard to V1 and L1 in visual brain and deep net, e.g., (3.2-3.5) / 3.5≈-8.) when considering progressively distant nodes or extra nodes. Negative numbers imply information loss and positive increments indicate a sort of synergy.

Secondly, we observed a general trend that, with increasing depth in both the visual brain and deep neural networks, there is a growing tendency to incorporate synergistic features (visual brain: $0 \rightarrow -8 \rightarrow -15 \rightarrow -1$ vs. deep nets: $0 \rightarrow 56 \rightarrow 67$) (refer to Table.4). This phenomenon suggests that as we delve into deeper layers, synergies become more prevalent, asserting their dominance in both biological visual processing and artificial neural networks Clauw et al. (2022; 2024). This parallels the notion that deeper levels of analysis and abstraction lead to a richer integration of complementary elements, fostering more complex representa-

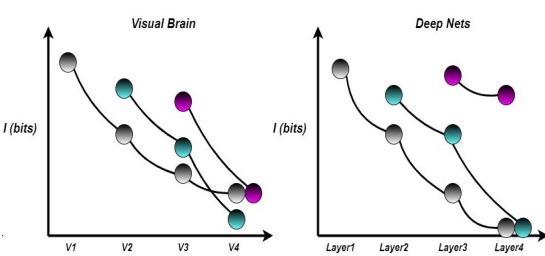

Figure 6: The data processing inequality in the visual brain and deep networks. From left to right, the figure illustrates how the DPI is satisfied in both the visual brain and deep networks. The colors indicate parallel information processing pathways.

tions Luppi et al. (2022); Wang et al. (2012). Thus, the deeper layers not only improve individual elements but also foster a smooth interaction that boosts overall perception and comprehension, whether in biological and deep neural networks.

## 4 DISCUSSION

The representation of access information in biological and artificial neural networks is a major pathway for us to understand how information changes during learning. Considering the interaction of multiple neurons in each layer of neural networks, traditional pairwise metrics may not be powerful enough to capture the statistical dependencies beyond pairwise relationships. Therefore, $TC$ were applied in this study to assess multi-way information interaction. From numerical and real experimental results, it was proven that $TC$ are superior to pairwise metrics. Subsequently, we applied them to solve real biological problems and discovered some very interesting issues. However, it still has some limitations that need to be mentioned here.

Firstly, there are several ways to assess high-order information interaction. In this study, we utilized $TC$, to evaluate high-order information interaction in both artificial and biological neural networks. However, given the importance of accuracy and precision in $TC$ estimations, it might be prudent to explore other methods for assessing high-order information interaction, such as persistence diagrams Yoo et al. (2016); Kumar et al. (2023), hypergraphs Battiston et al. (2020); Hanlin & Bianconi (2021), tensor decomposition Chatzichristos et al. (2018), and so forth. These alternative approaches could provide additional insights into explaining complex information interactions within complex systems.

Secondly, the way of dynamically representing information in deep neural networks is more complex than static situations. Accessing real-time information changes during learning is particularly intriguing for understanding the black box of deep nets. Moreover, the architecture and goals of deep nets are also major factors to consider when assessing information. Different architectures with distinct goals will undoubtedly lead to variations in information measures. In comparison to deep nets, accessing information in biological neural networks is becoming increasingly challenging Lindsay (2020). Precisely quantifying multi-way information interactions among neurons or brain regions has always been a challenging yet fascinating scientific endeavor. In this study, we measured multi-way information interaction in visual regions and uncovered some highly intriguing results, typically overlooked by pairwise metrics.

Thirdly, we suggested that DPI may holds in biological and artificial neural networks, and meanwhile, it also reflects the parallel information exchanges in neural networks, which explains the efficiency of neural networks. But to properly explain DPI in deep nets from a physics or other perspective is needed if we want to more deeply understand it. Unfortunately, we are still in the shallow layer for understanding DPI at both biological and non-biological neural networks, and here we thought that the meaning behind biophysics meaning of DPI would not just stay in the *efficient coding hypothesis*. However, it is important to note the injected noise strength: in our case, we added noise with an amplitude equal to 5% of the response amplitude in each layer when measuring information flow in both biological and artificial neural networks. This value, adopted from previous research Malo (2019), may not accurately reflect the actual noise levels present in real biological neural networks. Therefore, the results may vary if this noise level is adjusted.

Furthermore, we also noticed some similarities and differences between the visual brain and deep networks. Firstly, we see that redundancy dominates the shallow layers of the visual brain, and then synergy emerges in the deep layers of the visual brain. For deep networks, we noticed a general trend of increased synergy as the number of layers increased. But for deep nets, one constraint to examine is the applicability of this finding beyond AlexNet. While it's exciting to discover that deep layers display emergent synergies inside AlexNet, it will be interesting to validate this finding across different network topologies because different neural network topologies may exhibit unique properties and behaviors, and moreover, the specific dataset and task on which this observation is based should be considered, considering that the performance and behavior of neural networks can vary significantly depending on the dataset and task complexity Li et al. (2022a); Huang et al. (2024). While emerging synergies in deep layers may indicate effective information processing and representation learning, this does not always imply that shallow levels are completely redundant. Shallow layers may nonetheless play important roles in feature extraction and initial processing, even if they are

dominated by redundancies. Therefore, understanding the information change between redundancy and synergy will be very important to understanding feature processing in deep networks.

## 5 CONCLUSIONS

Multiway independence redundancy information was measured in both biological and deep neural networks using $TC$. Those approach provided us with a means to quantify multiway information interaction in biological visual systems and deep neural networks. In assessing multiway information representation in complex biological vision and deep nets, we first demonstrated the superiority of $TC$ over pairwise metrics. Subsequently, we applied them to real biological visual networks and deep neural networks. We observed that the data processing inequality holds true in both biological vision and deep nets, implying information parallel processing in the visual brain and deep nets and functional similarities between deep nets and the brain from an information-theoretical perspective. Furthermore, we observed some notable differences. In particular, we discovered that redundancy predominates in the early layers of biological vision, which contrasts with deep neural networks. However, synergistic information shows a general trend of gradually increasing in the deeper layers of both biological vision systems and deep neural networks. Interestingly, this mirrors findings from studies of the actual biological visual system.

### COMPLIANCE WITH ETHICAL STANDARDS

This research study was conducted retrospectively using human brain fMRI data obtained through open-access sources. Ethical approval was not required, as confirmed by the license accompanying the open access data.

### DECLARATION OF COMPETING INTEREST

The authors declare that they have no known competing financial interests or personal relationships that could have appeared to influence the work reported in this paper.

### USE OF LARGE LANGUAGE MODELS

In this work, the LLM is utilized only as an aid for refining and polishing the written text.

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

# A APPENDIX

## A.1 BACKGROUND ON ENTROPY AND MUTUAL INFORMATION

The Shannon entropy $h(X)$, or simply entropy, of a continuous random variable (RV) $X \in \mathcal{X}$ with probability density function $f_X$, its differential entropy is defined as Cover & Thomas (2006),

$$h(X) = - \int_{\boldsymbol{x} \in \mathcal{X}} f_X(\boldsymbol{x}) \log \left( f_X(\boldsymbol{x}) \right) dx$$

Then, the mutual information between $X$ and another continuous RV $Y \in \mathcal{Y}$ is given by Cover & Thomas (2006),

$$MI(X;Y) = h(X) - h(X \mid Y) = h(Y) - h(Y \mid X)$$

The mutual information measures the dependency between $X$ and $Y$, and attains its minimum, equal to zero, if they are independent. Alternatively, when considering $MI$, one can utilize the entropy of the Gaussian variables, yielding:

$$MI(X,Y) = -\frac{1}{2} log \left( 1 - \frac{\Sigma_{XY}^2}{\Sigma_X^2 \Sigma_Y^2} \right) = -\frac{1}{2} log \left( 1 - \rho^2 \right)$$

where $\Sigma$ is the covariance matrix, and $\rho$ is the correlation coefficient between $X$ and $Y$.

## A.2 REAL BIOLOGICAL VISION: V1,V2,V3,V4

To compare information flow in artificial neural networks with biological visual pathways, we utilized data from The Algonauts Project 2021 Challenge Cichy et al. (2021).

*Task paradigm:* Each of the ten participants completed five separate scanning sessions. The first scanning session consisted of a localizer experiment, where participants passively viewed short videos (distinct from the 1,102 test/train videos in the main experiment) belonging to various categories. The brain data obtained from this localizer experiment was used to define the location of the visual Regions of Interest (ROIs) in each subject. These ROIs span the ventral visual pathway from early and mid-level visual cortex (V1, V2, V3, and V4).

The remaining four sessions consisted of the main experiment and followed an identical structure. Participants were instructed to focus on a fixation cross at the center of the screen while they passively viewed the 3 second training and testing set videos without audio. The videos were presented in 13 separate runs, with each run lasting about 7 minutes and consisting only of either training videos or testing videos. By the end of the four main experiment sessions, each participant viewed each of the 1,000 training videos 3 times and each of the 102 testing videos 10 times.

*fMRI Preprocessing:* The data processing step includes slice time correction, realignment, co-registration, and normalization to MNI space. Additionally, the BOLD signal of each voxel in the preprocessed fMRI data was modeled as a weighted combination of simple Finite Impulse Response basis functions Lindquist et al. (2008). The estimated coefficients were averaged across time, resulting in a single averaged coefficient for each video presentation (in each voxel of each ROI). This response description is the one used as input data in our analysis.

## A.3 DEEP NEURAL NETWORKS

We utilized a pre-trained deep CNN known as *AlexNet* to extract hierarchical visual information from natural video stimuli. AlexNet Krizhevsky et al. (2012), trained on the Large Scale Visual Recognition Challenge 2012 dataset, consists of an 8-layer architecture: the first five layers are convolutional, and the last three are fully connected. Max-pooling is implemented between certain layers. The final layer utilizes a softmax function to classify input images into categories. Specifically, layers 6 through 8 contain 4096 units each, followed by 4096 units and then 1000 units, respectively (as shown in Fig.5). Once trained, a straightforward feedforward pass of an input video could extract features and conduct video recognition with the CNN. Activation occurred in each unit of the CNN when a natural video was fed into the system, representing the fluctuating representation of a specific feature in the video frame. An activation weight was generated from passing through

each frame. Units sharing the same kernel within a single layer output every frame as a feature map. The output of the rectified linear function before max-pooling in this section of the paper refers to each layer's output.

*Video Preprocessing:* The videos were fed into pre-trained deep neural networks, with 16 frames selected from each video, resulting in a total of $16 \times 1000$ samples. Subsequently, we extracted the response in each layer of the artificial neural networks. After saving the responses in each layer and addressing dimensioning issues and related computing efficiency, we read the corresponding responses and resampled them to achieve uniform spatial resolution ($6 \times 6$). Then, we concatenated all the responses in each layer, preparing to estimate the information measures in deep networks.

