# OpenReview forum: "Multiway Information Interaction Measures Reveal Redundant Coding in Biological and Artificial Neural Networks"
_ICLR.cc/2026/Conference — ICLR 2026 Conference Withdrawn Submission_

### Official Review · Reviewer_kibr · 2025-10-17

**Soundness:** 2
**Presentation:** 3
**Contribution:** 3
**Rating:** 2
**Confidence:** 4

**Summary:**

The paper proposes to measure total correlation (TC) in cortical and artificial neural networks to understand information processing from the perspective of redundancy. On a set of experiments with neurological data and ANNs, the paper draws conclusions about how redundancy and synergy change with the layer index. Comparisons with mutual information (MI) estimates suggest that multivariate information measures are preferable over bivariate ones, such as MI.

**Strengths:**

The paper treats an interesting topic. Indeed, information-theoretic analyses of (deep) learning systems are promising avenues to better understand neural networks, and papers on redundancy/synergy are still on the rise. Hence, the topic is absolutely timely. Furthermore, the inclusion of neurological data is a welcome addition.

**Weaknesses:**

I have several comments that currently prevent me from recommending acceptance:
- The paper only presents anecdotal evidence, as results are presented for only a single ANN and with only a single dataset. It is hence unclear if these results generalize to other architectures and tasks. While this anecdotal evidence is certainly interesting, I have the feeling that it does not meet the bar of a top-ranked conference yet.
- In parts, the conclusions seem to be conflicting. For example:
   - In Fig. 4 it is claimed that MI estimates are not reliable. At the same time, these MI estimates are used to show the DPI in Section 3.2.2. Does that mean that the results in Section 3.2.2 are unreliable?
   - The statement "we observe [...] in both the visual brain and DNNs [...] growing tendency to incorporate synergistic features" is in conflict with the numbers, which are negative for the visual brain but positive for the DNN. Did I misunderstand something?
- Some of the claims are not supported by sufficient evidence. For example,
   - With reference to Fig. 4, for MI no theoretical value is plotted. Hence, the conclusion that "the estimations from MI are incorrect and have less sensitivity to connectivity" is not supported. Indeed, MI and TC are different concepts, and it is clear that one could be more strongly influenced by connectivity than the other. It is important, though, to compare the estimators of MI and TC with their ground truth value to see if they can be reliably estimated. Whether one _should_ estimate MI or TC is a different question, but that question can not be answered by comparing estimators only to the theoretical values of TC.
   - Any statement about synergy is unsupported. Synergy is not the opposite of redundancy, indeed, partial information decomposition considers redundant, synergistic and unique information. A reduction of redundancy can be due an increase of synergy or an increase of unique information.
   - The statement "deeper layers [...] foster a smooth interaction that boosts overall perception" is vague (what does smooth interaction mean? what does boost perception mean?) and speculative. If at all, it better fits inside the discussion section.
- The mathematics are not always clear. For example:
  - (3) could be derived in some appendix, or the reference to Laparra et al. could be more concrete, pointing to a theorem or an equation there. Similarly, the same holds for the offset equation  ($TC= C+J-J_m$) between (6) and (7). The terms $J$ and $J_m$ should also be defined in equations, and not only in text. In (7), the sum apparently runs over both logarithms, so brackets are missing.
   - An important missing investigation is how the TC-RBIG estimate is affected by the dimensionality of the latent representation. Total correlation in itself is affected by the dimensionality of the data (at least under the assumption of almost lossless transformations). In addition, it would be important to study how the estimate of TC is affected by dimensionality. I understand that the dimensionalities used in Fig. 4 are quite small compared to those in the experiments, so a sensitivity analysis on synthetic data (where the true TC is known) is essential.
   - The experimental setup, i.e., how TC estimates are calculated precisely, is missing. As such, the experiments are not reproducible from the text.
- The writing style is sloppy at some occasions and requires revisions (e.g., lines 256-262).

Minor points:
- In Fig. 1 it is not clear how subfigure B is related to TC, as claimed in the caption.
- The abbreviation DNN is introduced but only rarely used in the paper.
- The text around eq. (1) apparently assumes continuous RVs/distributions, but this is not stated explicitly.
- In (4) a line break seems to be missing.
- log should be replaced by \log.
- In Fig. 4, the legend entry "I-RBIG" is not clear.
- citet is consistently used throughout the paper, while in several occasions citep is more appropriate.

**Questions:**

Please mainly respond to the weaknesses above, which have some implicit questions. For the sake of filling this mandatory form, I summarize the main questions here:
- How can the results of Sec. 3.2.2 be justified considering your claims around Fig. 4?
- How does the statement "we observe [...] in both the visual brain and DNNs [...] growing tendency to incorporate synergistic features" fit with the numbers reported next to it?
- In Fig. 4, what is the theoretical value of MI?
- What are the formulas for $J$, $J_m$?
- How is the TC estimate affected by the dimensionality of $X$?
- How are TC values estimated in detail? Please provide some reproducible description.

---

### Official Review · Reviewer_Zdku · 2025-10-31

**Soundness:** 2
**Presentation:** 1
**Contribution:** 2
**Rating:** 2
**Confidence:** 3

**Summary:**

This paper investigates multiway information interactions in both biological and artificial neural networks, making use of information-theoretic measures—specifically, total correlation (TC)—to quantify higher-order dependencies and redundant coding. The authors present both theoretical and empirical analyses, comparing the sensitivity of pairwise metrics (mutual information, MI) versus TC, and apply these measures to fMRI data from human visual cortex as well as activations in AlexNet, a classical deep convolutional neural network. Results indicate that TC captures high-order redundancy overlooked by MI, provides evidence for the data processing inequality (DPI) in both biological and artificial processing pathways, and reveals differing redundancy/synergy trends across layers in brains and deep networks. The work aims to bridge neuroscience and deep learning perspectives through advanced information analysis.

**Strengths:**

**1.** The research task is important, and the application of correlation-based measures to modern machine learning problems is both timely and relevant.

**Weaknesses:**

**1.** The Method section mainly reiterates existing techniques without introducing any original methodological contribution. This significantly diminishes the paper’s overall novelty and makes the work appear more as an application or synthesis rather than a genuine research advance.

**2.** The content is relatively limited, and the paper suffers from several layout and presentation issues. Some figures are confusing or poorly explained, and the mathematical notation contains inconsistencies and typographical errors. These problems reduce readability and make it difficult to fully assess the technical quality of the work.

**3.** Experimental Scope and Generality: The empirical analysis is limited to AlexNet as a representative deep network. However, AlexNet is now considered outmoded in deep learning, and no evidence is given that these redundancy/synergy trends generalize beyond this architecture, task (video classification), or dataset.

**Questions:**

NA

---

### Official Review · Reviewer_Veo4 · 2025-11-01

**Soundness:** 3
**Presentation:** 2
**Contribution:** 2
**Rating:** 4
**Confidence:** 2

**Summary:**

This paper proposes using total correlation rather than previously-in-use pairwise metrics for assessing the mutual information between the representations at different layers of both neural networks and the brain. It splits its time between describing and arguing for this information measure, and then demonstrating the usage of it for both biological and ANN data.

**Strengths:**

- I appreciated the RBIG method, and the way it used a multi-step process that could be understood at each level in the chain, to capture a total information loss between the baseline distribution and the fully whitened Gaussian

**Weaknesses:**

- The paper dived into an explanation of the details of Total Correlation without giving a good grounding of what exactly it meant to do mutual information analysis on both ANN and neural data, leaving me confused about the actual proposed application even as there was a lot of detail given of the information theoretic aspects of the approach
- This paper assumed a very deep understanding of specifically pairwise information theoretic measures, and did nothing to make any of their points or methods comprehensible to someone without that specific background. The paper was confusing and overwhelming as a result

**Questions:**

- Are there particular kinds of network architectures that your model would or wouldn't work on?
- Suggestion: this paper would benefit from a presentation overhaul that tried to clarify its intent, subject area, and concrete intended application early on

---

### Official Review · Reviewer_GVvC · 2025-11-04

**Soundness:** 1
**Presentation:** 1
**Contribution:** 1
**Rating:** 0
**Confidence:** 3

**Summary:**

The authors seek to assess the amount of higher-order interactions between both regions of the visual cortex as well as between interactions between layers in convolutional neural networks.  The authors apply a recent estimation technique for total correlation to fMRI data and use activation values for AlexNet.  The authors show that important properties, such as the data processing inequality, appear to hold.

**Strengths:**

- The broad topics this paper touches on of (1) higher-order interactions in the brain, (2) information processing in deep neural networks, (3) and estimation methods for information theoretic quantities are all great topics relevant to ICLR and the broader ML/AI community.
- The questions (at a general level) the authors ask, such as whether the human brain’s visual processing is a chain network or more complex and the amounts of redundancy/synergy in the brain (or deep neural nets) are also important

**Weaknesses:**

Although the broad topics and questions posed are quite interesting, there does not appear to be any contribution technically nor thorough (i.e. more than just one data set) empirical investigation.

## Major
- Main contributions (end of Section 1)
    - the authors claim as contribution 1 that they “Introduced and applied beyond-pairwise information-theoretic metrics (e.g., total correlation) to quantify high-order interactions in both biological and deep neural networks”  Others have done this (‘introduced and applied’);  the authors cite two papers with Greg Ver Steeg for instance involving functional connectivity using the total correlation, for instance.  Laparra et al. (2025) who proposed the RBIG estimator used by the authors also applied their total correlation RBIG estimator to visual area data.
    - Claimed contribution 2 “Demonstrated the limitations of pairwise metrics by showing their insensitivity to capturing high-order dependencies” that pairwise values can be misleading is straightforward
    - contribution 3 “Revealed functional parallels and distinctions between deep neural networks and the visual brain through high-order redundancy analysis, offering new insights into their internal representations.”  My understanding was that the parallels were not functional (that may have been hinted through the set up and presentation).  There was a single data set used and no discussion on findings from the neuroscience community were discussed.


- Section 2 “THEORETICAL ANALYSIS” appears to be entirely background material
    - 2.1 is total correlation (from (Watanabe, 1960))
    - 2.2 is a total correlation estimator from Laparra et al. (2025)
    - 2.3 is just stating the classic data processing inequality.

- Section 3.1 appears to be two pages for a single toy synthetic system, where the main conclusion is that total correlation is different than Pearson correlation.

- Section 3.2.2
    - a feedforward CNN will satisfy the DPI by its design.  It is unclear why this is measured, unless the intention is to draw connections (not discussed in the section) about how the values might relate between the two tables.
    - the question of how information flows in the visual cortex has been studied by neuroscientists for decades but there is no discussion of even classic let alone recent neuroscience findings on the human visual system.
    -  I do not understand the point of using the same stimuli for both the CNN as was used to collect the fMRI data for the point of assessing DPI.
- ‘redundancy’ in particular and ‘synergy’ as well are referred to throughout the paper, but I don’t think were formally/clearly defined.
    - Line 362 mentions that for Table 4 “positive increments indicate a sort of synergy.”
- Claims made are too strong for the level of exploration carried out.  Line 388 “From numerical and real experimental results, it was proven that TC are superior to pairwise metrics. Subsequently, we applied them to solve real biological problems and discovered some very interesting issues.”
## Minor
- lines 057-059 “our primary objectives include quantifying redundant information in both t the brain and DNNs,”  is far more ambitious of a objective than the work done supports.
- lines 61-62 what is “independent redundancy”?
- The figures need revision – several are low resolution (esp Fig 4 and Fig 5), have fonts that do not match main paper font sizing
- Fig 1
    - I do not understand what is being shown.  There is clearly visual distinctions between each element, but eg what do Venn diagrams, which show overlap between sets, depict higher-order interactions (namely, the figure should make it clear what the sets are such that the intersection corresponds to something to be conveyed)
    - for (A), why are neuron interactions depicted as undirected (hyper) edges?  Aren’t edges directed in biological and artificial neural networks?

- Fig 5
    - Sizing/resolution issues
    - There are too many different parts to this image, I find it challenging to read/understand.  At the least this could be broken up into parts with better labeling.
    - is the depiction of the human supposed to show an analog of the CNN architecture?  If so, that is not clear.
   - what do the numbers in the boxes of “Pairwise” and “Higher-Order Triadic” information mean?  Is the point of that just to show there are many more possible triads than pairs (105^2 pairs and 105^3 triplets)
    - for those boxes, shouldn’t it be $n \choose k$ instead of $n^k$? eg for pairwise there is no notion of order (equivalently mutual information is symmetric in its argument).   Or is the ordered counting supposed to be based on directed edges (that say the bottom left subfigure with edges T1 $\to$ T2 and vice versa are ordered, and you want to emphasize I(T1:T2) captures both?  If so, then how the triadic enumerations (6 orderings of {T1, T2, T3}) correspond to the colorings of the edges in the figure in the bottom row is unclear to me.

## Very Minor
- the authors use in text citations throughout in places parenthetical citations should be used.  eg linke 037 should be “… brain functions (Wang et al., 2023; Li et al. …)”
- Fig 1 text in the image font size should be increased
- line 257 “is way more appropriate”
- line 316 “Considering that the internal representation may contain some amount of noise,” which of the systems is that referring to? AlexNet?
- line 385 “The representation of access information”
- line 388 “Therefore, TC were applied”

**Questions:**

N/A

---

### Note · Authors · 2025-11-17

**Comment:**

Dear Area Chairs, Program Chairs, and Reviewers,

We would like to respectfully request the withdrawal of our submission from the ICLR review process.

After carefully reading the reviews and reflecting on the constructive feedback and scores provided by the reviewers, we realized that several aspects of the current manuscript, particularly regarding clarity, novelty, and the need for additional empirical evaluation, would benefit from significant revision. We feel that undertaking these improvements properly would require changes beyond what is feasible within the remaining review cycle.

In light of this, we believe it is more appropriate to withdraw the paper at this time so that we can thoroughly address the concerns and strengthen the overall contribution. We deeply appreciate the reviewers’ thoughtful and detailed comments; they have provided valuable guidance that will greatly help us improve the work for a future submission.

Thank you very much for your understanding and for the time and effort invested by the reviewers and AC team.

With sincere appreciation,

on behalf of all co-authors

**Withdrawal Confirmation:**

I have read and agree with the venue's withdrawal policy on behalf of myself and my co-authors.